# Roof-Inhabiting Cousins of Rock-Inhabiting Fungi: Novel Melanized Microcolonial Fungal Species from Photocatalytically Reactive Subaerial Surfaces

**DOI:** 10.3390/life8030030

**Published:** 2018-07-15

**Authors:** Constantino Ruibal, Laura Selbmann, Serap Avci, Pedro M. Martin-Sanchez, Anna A. Gorbushina

**Affiliations:** 1Department of Pharmacology, Pharmacognosy, and Botany, Faculty of Pharmacy, Universidad Complutense de Madrid, 28040 Madrid, Spain; tinoruibal@yahoo.com; 2Department of Ecological and Biological Sciences (DEB), University of Tuscia, 01100 Viterbo, Italy; selbmann@unitus.it; 3Italian National Antarctic Museum, Mycological Section, 16128 Genoa, Italy; 4Department 4 Materials and Environment, Bundesanstalt für Materialforschung und -prüfung (BAM), 12205 Berlin, Germany; aag1936@gmail.com; 5Section for Genetics and Evolutionary Biology (EVOGENE), Department of Biosciences, University of Oslo, 0371 Oslo, Norway; p.m.martin-sanchez@ibv.uio.no; 6Department of Earth Sciences & Department of Biology, Chemistry, Pharmacy, Freie Universität Berlin, 14195 Berlin, Germany

**Keywords:** microcolonial fungi, multilocus phylogeny, photocatalytic surfaces, subaerial biofilms, stress tolerance, *Constantinomyces*

## Abstract

Subaerial biofilms (SAB) are an important factor in weathering, biofouling, and biodeterioration of bare rocks, building materials, and solar panel surfaces. The realm of SAB is continually widened by modern materials, and the settlers on these exposed solid surfaces always include melanized, stress-tolerant microcolonial ascomycetes. After their first discovery on desert rock surfaces, these melanized chaetothyrialean and dothidealean ascomycetes have been found on Mediterranean monuments after biocidal treatments, Antarctic rocks and solar panels. New man-made modifications of surfaces (e.g., treatment with biocides or photocatalytically active layers) accommodate the exceptional stress-tolerance of microcolonial fungi and thus further select for this well-protected ecological group. Melanized fungal strains were isolated from a microbial community that developed on highly photocatalytic roof tiles after a long-term environmental exposure in a maritime-influenced region in northwestern Germany. Four of the isolated strains are described here as a novel species, *Constantinomyces oldenburgensis*, based on multilocus ITS, LSU, RPB2 gene phylogeny. Their closest relative is a still-unnamed rock-inhabiting strain TRN431, here described as *C. patonensis*. Both species cluster in Capnodiales, among typical melanized microcolonial rock fungi from different stress habitats, including Antarctica. These novel strains flourish in hostile conditions of highly oxidizing material surfaces, and shall be used in reference procedures in material testing.

## 1. Introduction

Melanized, slow-growing and stress-tolerant microcolonial ascomycetes are considered the most persistent inhabitants of bare rock surfaces [1,2]. After their first discovery on desert rocks covered by rock varnish [3], melanized ascomycetes from Chaetothyriales and Dothideomycetidae have been found on Mediterranean monuments and natural rock outcrops (e.g., [4,5,6,7,8,9]), Acropolis marble after biocidal treatments [10], Antarctic rocks [11,12] and solar panels [13]. Atmosphere-exposed surfaces, as hostile environments open to sun irradiation and fluctuating water and nutrient availability, offer themselves as a perfect niche selecting for these survival specialists.

Manmade materials like building surfaces, monuments, solar panels, etc. have to be protected from the growth of these organisms that might cause pitting [6], color modification [14,15,16] and physical damage of the underlying substrate [17]. Therefore, diverse microbial control technologies are employed to protect these surfaces by modifications such as biocides as well as surface treatments with photocatalytically active layers. These growth-controlling amendments might, however, lead to development of resistance in SAB and thus further select for stress-tolerant microorganisms, including melanized microcolonial fungi.

This investigation was based on a roof-tile exposure case study in Northwestern Germany (Edewecht). Several melanized microcolonial isolates were obtained from tiles with a photocatalytically active surface that were exposed from November 2006 to May 2014. Four of them have been morphologically characterized and here described as a new species in the genus *Constantinomyces* based on a three-gene multilocus phylogeny that includes both melanized rock-inhabiting and plant pathogenic capnodialean fungi.

## 2. Materials and Methods

### 2.1. Sampling and Isolation

Roof tiles covering a residential house were exposed at an angle of 45° in a northerly direction. Exposure started in November 2006, and conventional clay tiles of traditional red color but with different types of surface modifications were juxtaposed. Material with photocatalytic surface properties covered part of the roof while the remaining roof surface comprised unmodified conventional clay tiles. Both tile types formed complete rows from the roof top to the edge.

The roof surface and the growth/soiling development on it was observed with documentation by photography from 2006 to 2014. Unfortunately, the specific roof topography influenced sun irradiation quality and duration, therefore reducing the photocatalytic activity of the surface layer. To concentrate our study on differences in colonization between tiles with different surface qualities, a direct comparison was ensured by placing photocatalytically active tiles among the control ones.

Two tiles with surface modifications were exposed from 2006 to 2009 in a shaded area of the roof. After 2.5 years of exposure (in July 2009), these two tiles with photocatalytic surfaces were placed on the control part of the roof (Figure 1a) where shading by architectural elements could be completely avoided. Thus for the period from July 2009 to May 2014 environmental conditions for the control and photocatalytically active tiles remained exactly the same. All sampled tiles were therefore subjected to semistandardized and reciprocally comparable conditions of a typical subaerial exposure in the maritime-influenced, moderate climate for an extended period of almost five years. The sampling was performed in May 2014 after 7.5 years of total exposure, and concentrated on three tiles—two with photocatalytically active surface (Figure 1) and one control.

Surface colonization phenomena were documented by photography prior to removing fragments to Petri dishes for subsequent cultivation. Photocatalytically active tiles have obviously different subaerial biofilm that is more discoloring (Figure 1a). Both sampled photocatalytically active tiles demonstrate an expressed subaerial biofilm growth in the form of dark pigmented fungal colonies (Figure 1b). Especially prominent were the colonies on the elevated surface area—where time of wetness is additionally reduced, leading to less pronounced algal growth.

The sampling goal was to isolate fungal strains from the visible dark colonies (Figure 1b) that were obviously dominating the atmosphere-exposed surface of the material. Fungal colonies that were sampled with sterile implements to isolate dominant subaerial settlers are indicated by the blue frame in Figure 1b.

Several developed colonies from the surface (Figure 1b) were sampled with sterile implements and parts of them were directly transferred to agar media [5]. The same procedure was also applied to the adjacent control tiles. The material removed from the surface and used for obtaining strains was also directly microscopically observed and only dark-pigmented fungal cell clusters were detected in situ, with admixture of green algae in lower-situated tile areas. The resulting melanized microcolonial strains were isolated and subcultured on malt extract agar (MEA) and subsequently stored at 4 °C.

### 2.2. DNA Extraction, PCR Amplification, and DNA Sequencing

Genomic DNA of each strain was extracted from its mycelia grown on MEA [13]. Mycelia were harvested and transferred to a 2-mL Eppendorf tube containing 500 μL Tris-NaCl-EDTA (TNE) buffer (10 mM Tris-HCl, 100 mM NaCl, 1 mM EDTA; pH 8) and glass beads of three different diameters (0.5, 2 and 5 mm). The mixture was shaken at 4.5 m/s for 1 min in a Fast Prep RiboLyser cell disrupter (Thermo Hybaid GmbH, Heidelberg, Germany). The DNA was purified through phenol/chloroform extraction and ethanol precipitation. The extracted DNA was resuspended in 100 µL sterile ultrapure water, and DNA concentration was quantified using NanoDrop 2000C (Thermo Fisher Scientific, Wilmington, DE, USA), following the manufacturer’s instructions.

Three molecular markers were amplified by conventional PCR for all fungal strains, the rDNA regions comprising ITS1, 5.8S rRNA gene, ITS2 and partial 28S rRNA gene (ITS + LSU), using the universal primers ITS5 [18] and LR5 [19], and a portion of the RNA polymerase II’s second largest subunit (RPB2) gene using the primers fRPB2-5f and fRPB2-7cr [20]. PCR reactions were performed in a BioRad C1000 Thermal Cycler (BioRad, Hercules, CA, USA). Cycling parameters for ITS+28S were 95 °C for 5 min, followed by 35 cycles of 95 °C for 30 s, 50 °C for 30 s, and 72 °C for 2 min, with a final extension at 72 °C for 10 min. For RPB2, cycling parameters were 95 °C for 5 min, followed by 35 cycles of 95 °C for 1 min, 55 °C for 2 min, and 72 °C for 1.5 min, with a final step at 72 °C for 10 min. Reactions were performed in duplicate, and negative controls (containing no DNA) were included in each PCR trial. All PCR products were checked by electrophoresis in 1.5% (*w*/*v*) agarose gels stained with GelRed dye (Genaxxon Bioscience GmbH, Ulm, Germany) and visualized under UV light. Positive amplification products were purified by using a QIAquick PCR Purification Kit (Qiagen GmBH, Hilden, Germany) and sequenced in duplicate by Macrogen Europe Company (Amsterdam, The Netherlands) with the same primer sets used for PCR. After edition, final sequences were submitted to the European Nucleotide Archive (ENA, EMBL-EBI) under the accession numbers listed in Table 1.

### 2.3. Alignment and Tree Reconstruction

ITS, LSU and RPB2 nucleotide sequences were compared in GenBank using BLASTN search [21], and RPB2 protein sequences were additionally checked using BLASTP; both algorithms were run through the NCBI website. The most similar sequences, as well as reference sequences from a selection of rock-inhabiting and plant pathogenic fungi in the order Capnodiales, were exported and iteratively aligned using the MUSCLE option [22] in MEGA6 [23]; final alignment was improved manually. The one-gene alignments were concatenated using FABOX [24] to combine sequences for the three-loci dataset. Optimized alignment was exported and the best-fit substitution model was determined using MODELTEST MrAIC 1.4.3 [25] as implemented in PHYML [26]. The phylogenetic tree was reconstructed by the maximum likelihood method, using TREEFINDER [27]. The robustness of the phylogenetic inference was estimated using a bootstrap method [28] with 1000 pseudoreplicates generated and analyzed with TREEFINDER.

### 2.4. Morphological Characterization

Microscopic features were characterized after 12 and 35 weeks of growth on 4% MEA incubated at 15°C. Mycelium and other fungal structures were then transferred onto slides mounted in 5% KOH and observed with light microscopy. Digital images were acquired using an Eclipse 80i microscope equipped with a DS-Fi1 digital camera and a Digital Sight DS-L2 image acquisition system (Nikon, Badhoevedrop, the Netherlands). Colony descriptions are based on cultures grown onto 4% MEA incubated at 15°C for 35 weeks.

## 3. Results

During the period of 2006–2014, a whole series of changes occurred on the air-exposed (subaerial) surface of the investigated roof. Alterations in the material surface color were the most apparent signs of the environmental exposure and accompanying colonization, and attracted our attention to a prominent difference between the photocatalytically active and control roof tiles (Figure 1). These color changes result from a combination of airborne deposition and biogenic conditioning by the subaerial biofilms. In this study, we aimed at characterizing the organisms forming subaerial biofilms on these surfaces.

### 3.1. Subaerial Biofilm Development and Isolation

The biofilms on the experimental roof developed through different phases: the first contrast was achieved in 2008, when the photocatalytically active part was seemingly free from the fouling biofilm, while the conventional tiles were overgrown with an intensely green-colored biofilm (results not shown). It is interesting to mention that from 2006 until 2009, photocatalytically active tiles were visually biofilm-free, while the control tiles just a couple of m away were covered in biofilm (data not shown). Exactly at that moment, two tiles with a photocatalytically active surface not visibly overgrown by a biofilm were transferred to the unshaded control part of the roof. Through the transfer of the photocatalytically active tiles, the following goals were accomplished: (i) any shading was avoided, thus ensuring the same environmental exposure for all tiles under study and (ii) the only difference influencing colonization was the type of surface.

In the following years, the colonization of tiles changed; two photocatalytically active tiles after almost five years of exposure demonstrated clearly different and visible biofilm growth (Figure 1b). Subaerial biofilms formed by melanized microcolonial fungi and some green algae were covering the sample as well as all other photocatalytically active tiles. Also, with the help of light microscopy of surface swabs material, only dark-pigmented fungal cell clusters were observed in situ, with admixture of the green algae in lower-situated tile areas. In May 2014, these biofilms were intensively colored and dominated by the dark fungal colonies, which were frequently single and especially prominent on the elevated areas of the roof tiles where time of wetness is additionally reduced (Figure 1b).

As we were mainly interested in organisms capable of withstanding the stress of subaerial exposure with the additional stress of photocatalytic oxidation, we sampled selectively the strains that were developing visible colonies on the material surface. Though the same cultivation procedures were applied to the photocatalytic tiles as well as control tiles, isolation from control tiles did not deliver any melanized microcolonial strains.

### 3.2. Strains Analyzed

Melanized microcolonial strains T2.1, T2.3, T2.4 and T2.5 were analyzed and compared to strain TRN431 (Table 1). Phylogenetic analyses conducted in this study showed that the four strains isolated from photocatalytically active roof tiles represent a taxonomic novelty within the class Dothideomycetes, order Capnodiales. As is frequently the case with microcolonial melanized ascomycetes, the observed strains were scarcely differentiated from a morphological point of view, and therefore taxonomic decisions were based primarily on partial sequences of nucLSU, RPB2, and on the complete ITS regions of rDNA. 

### 3.3. Sequences and BLAST Search

Ribosomal sequences (complete ITS plus partial LSU) obtained for the four strains from roof tiles were 1382-4 bp length. Their sequence variability was rather limited but with a few differences located in 10 nucleotide positions, which mostly support the separation of two groups of strains (T2.1/T2.5 and T2.3/T2.4). All four strains showed the same closest related sequence using BLASTN search: *Teratosphaeria* sp. CPC13917 (EU707885; 94% similarity) isolated from *Protea nitida* leaves, followed by diverse *Colletogloeopsis* and *Teratosphaeria* species (93%).

The partial RPB2 sequences were 1037 bp length. RPB2 sequences of strains T2.1 and T2.5 were identical, but significantly different than T2.3 (93.2% similarity) and T2.4 (92.3%). Likewise, the similarity between T2. 3 and T2.4 was 93.2%. However, after protein translation, the four strains showed identical sequences of amino acids. With regard to BLAST results, using nucleotide DNA sequences (BLASTN) the four strains showed the same closest related sequence *Davidiellaceae* sp. CBS117950 (GU371755; 90%); using protein sequences (BLASTP), the best hits were *Davidiellaceae* sp. CBS117950 (ADB93966; 99% aa) and *Teratosphaeriaceae* sp. CBS117937 (ADB93965; 90% aa), both rock isolates.

### 3.4. Alignment and Tree Reconstruction

Phylogeny was based on the concatenated 3-genes dataset that included 59 strains in *Teratosphaeriaceae* whose sequences could be confidently aligned with sequences of the studied strains; the combined dataset was based on 1681 positions (ITS: 1–525; LSU: 526–1408; RPB2: 1409–1681) including gaps. The specimens T2.1, T2.3, T2.4, T2.5 here analyzed shared five positions in ITS sequences that were different in the other close relatives analyzed; the positions and substitutions in the analyzed specimens/closest relatives are reported as follows: position 26: T/C; position 30: T/C; position 54: T/A or C; position 171–172: TT/AC or AA or AT or TC. The multilocus phylogenetic tree was generated using a GTR + G(4) model selected using the Akaike’s information criterion with a maximum likelihood (ML) approach. The tree was rooted with the strains *Cladosporium cladosporioides* CBS112388 and *Cladosporium herbarum* CBS121621 (Cladosporiaceae) and is shown in Figure 2.

The sequences analyzed belonged to strains of both rock-inhabiting and plant pathogenic fungi and also some known extremophiles, such as the halophilic species *Hortaea werneckii*; all strains analyzed sit in the *Teratosphaeriaceae*, order Capnodiales. The base frequencies were as follows: T = 0.2163, C = 0.2628, A = 0.2320, G = 0.2887.

The topology of the tree and positions of the groups, even if based on a much more restricted selection to focus on the genus *Constantinomyces* and relatives, is in accordance with wider recent phylogenetic studies on the family *Teratosphaeriaceae* [29]. Although the species *C. nebulosus* was placed far from *C. macerans* and in a misguided position in the reference study, all the species hitherto described in the genus *Constantinomyces* cluster together in our phylogeny. All genera highlighted in the tree were well separated and, including the genus *Constantinomyces* supported with 90% bootstrap. The four isolates from photocatalytic surfaces, here described as *C. oldenburgensis*, pooled together in a cluster with 99.7% bootstrap support; the rock-inhabitant strain TRN431 was clearly outside this group, rather isolated on a longer branch. Based on these phylogenetic evidences and different ecology, we have described this strain as a separate new species *C. patonensis*.

### 3.5. Taxonomy of *Constantinomyces oldenburgensis* sp. nov. and *Constantinomyces patonensis* sp. nov.

Phylogenetic analyses conducted in this study showed that the strains isolated from photocatalytically active roof tiles represent a taxonomic novelty within the class Dothideomycetes, order Capnodiales. The observed strains were morphologically scarcely differentiated from a rock-surface isolate, TRN431, from the Iberian penninsula. Therefore taxonomic decisions were based primarily on partial sequences of nucLSU, RPB2, and on the complete ITS regions of rDNA.

Descriptions based on 35-week-old cultures grown on MEA at 15 °C.

***Constantinomyces oldenburgensis*** Gorbushina, Martin-Sanchez, Selbmann & Ruibal, **sp. nov**. MycoBank MB825220, Figure 3a–g.

Holotypus: CBS144642 = T2.1 = CCFEE6311, from a photocatalytically active roof tile, Edewecht, Germany. Culture preserved in liquid nitrogen and in dried condition.

Etymology: named after Oldenburg, the German city near the isolation source where the work on black fungi from monuments and building materials started in the 1990s.

Diagnosis: Descriptions based on 35-week-old cultures grown on MEA at 15 °C. Colonies growing very slowly in irregularly shaped colonies, velvety dark olive green to dark brown to black, dark in reverse, with regular margin, sometimes developing radiated, submerged margin at long cultivation times, raised centrally, flat near the margin. Hyphae pale to dark brown, septate, thick-walled, with apical germination producing elongated, cylindrical, irregular hyphae; in later stages torulose hyphae often present at sections, composed of swollen cells with or without transverse septa, brown, thick-walled, smooth. Chlamydospore-like cells sometimes present. Dark, spherical multicellular bodies sometimes present. Conidiophores micronematous, conidiogenous cells intercalary, occasionally with thallic-arthric conidia, liberating sparcely. Teleomorph unknown.

Note: Arthroconidia are generated by hyphae fragmentation but secession is not always completed, and adjacent cells remain connected by intercalary elongations of narrow pale connectives. Multicellular bodies are typical of the species *C. oldenburgensis* since they have not been reported in any of the other hiterto described species in the genus (*C. virgultus, C. macerans; C. minimus; C. nebulosus*). A similar trend in hyphal progression was observed in *Elasticomyces elasticus*, another globally distributed microcolonial black fungus developing on rocks.

***Constantinomyces patonensis*** Ruibal & Selbmann, **sp. nov**. MycoBank MB825222, Figure 3h–k.

Holotypus: CBS117950 = TRN431, from granite, Patones, Central Mountain System. Culture preserved in liquid nitrogen and in dried condition.

Etymology: named after Patones, the locality in the Spanish Mountain System where the strain was isolated.

Diagnosis: Descriptions based on 35-week-old cultures grown on MEA at 15 °C. Colonies growing very slowly in irregularly shaped colonies, velvety dark olive green to dark brown to black, dark in reverse, with regular margin, raised centrally, flat near the margin. Hyphae pale to dark brown, septate, thick-walled, with apical germination producing elongated, cylindrical, regular hyphae; in later stages torulose hyphae often present at sections, composed of swollen cells with or without transverse septa, brown, thick-walled, smooth. Teleomorph unknown.

Note: No conidiogenesis of any type has been observed for this species. None of the structures observed in *C. oldenburgensis*, such as multicellular bodies or chlamydospore-like cells, have been found in *C. patonensis*. All these differences, together with those regarding its biogeography and mainly the molecular divergence of its nucITS DNA fragment, account for its definition as a different species from *C. oldenburgensis*.

## 4. Discussion

Microcolonial black fungi are specialized in exploiting different stressing niches, from saltpans to acidic sites, from hot deserts to Antarctic rocks [7,8,9,30,31,32,33,34]. They show a stunning ability to resist and overcome a number of different challenges as high and low temperatures, drought, osmotic stress, high solar and UV radiation or prolonged period of water deficiency [32,35,36,37]. Among other characteristics, the presence of melanins in the cell wall enable them to cope with different stressors, including UV- and quite intense ionizing radiations [38].

The working hypothesis in this study was that microcolonial melanized ascomycetes, usually forming resistant biofilms on subaerial rock and material surfaces, flourish in hostile conditions of highly oxidizing material surfaces and that, under these conditions, the occurrence of novel species is promoted. We also hypothesize that in additionally stressed systems that are designed for growth avoidance—like on photocatalytically active surfaces and in biocide-treated materials—stress-tolerant microcolonial ascomycetes are going to be dominant.

Four presented isolates of *Constantinomyces oldenburgensis* were components of a developed subaerial biofilm that formed in 7.5 years under the influence of normal environmental exposure. Growth of these material-inhabiting melanized ascomycetes was not inhibited on photocatalytically active surfaces of environmentally exposed roof tiles. It is even possible that the additional stress of photocatalytic oxidation selects for melanized, stress-tolerant strains, because (i) no microcolonial isolates could be obtained from the control tile surfaces and (ii) subaerial biofilms on the control tiles were significantly less pigmented (Figure 1a). Even if we assume that photocatalysis is not exceptionally active during mild cloudy winters of Northern Germany, differences between the control and photocatalytic surface were sufficient to ensure that microcolonial black fungi were prevailing isolates only on roof-tile surfaces with photocatalytic oxidation stress.

The genus *Constantinomyces* includes typical rock-inhabiting fungi, possessing two important qualities (i) occurrence on bare solid surfaces in the Mediterranean area and (ii) characteristic morphology and properties of microcolonial surface-inhabiting fungi. Ubiquitous findings of these strains underline the similarities existing between SAB niches on man-made materials (that are recent anthropogenic additions to Earth ecosystems) to naturally existing rock surfaces that were the first terrestrial environments of this planet [1].

The intraspecific variability within the *C. oldenburgensis* clade accounted to 1.5%. This is not as high as found for other black microcolonial fungi, as *Friedmanniomyces endolithicus* or *Elasticomyces elasticus*, where the intraspecific variability was calculated as high as 4–4.5%. This may be explained as a sampling effect, since the cluster here analyzed is composed of four strains only, while the two examples reported above included up to 10 representatives [31]. Alternatively it could be a consequence of the restricted sampling location: in fact the strains here examined came from photocatalitically active surfaces of two adjacent tiles (Figure 1d), while strains of the examples above were from distant locations (for *E. elasticus* even different continents). Based on the phylogenetic position of TRN431 (rather isolated compared to all the other species, on a long branch and support of the common branch 98.6%) as well as considering its different ecology (*C. oldenburgensis* is from roof tiles and TRN431 from natural rocks) we described TRN431 as a separate species, *C. patonensis*.

Melanized rock- and material-inhabiting (essentially subaerial-biofilm-inhabiting) ascomycetes form a resource of climate-dependent and anthropogenically influenced biodiversity. Characteristic properties of melanized rock-inhabiting fungi are their stress tolerance and substrate interaction abilities. These aptitudes enable these organisms to form visible and biodeterioration-involved subaerial biofilms on exposed modern materials—exactly in the same fashion as their ancestors once conquered the surfaces of desert boulders. As reactions of these melanized microcolonial fungi to growth-avoiding measures will give us clues to design material protection measures, these organisms are a focus of applied research. Melanized fungi that persist under realistic material exposure conditions are also promising candidates as new reference organisms in material testing procedures.

## 5. Conclusions

Progress in biodeterioration sciences depends on reflecting the constantly evolving biodiversity of the biofilms associated with specific biodeterioration phenomena. The diversity on bare material surfaces is high, largely unexplored and highly dynamic. Forces that influence this biodiversity are the introduction of new materials along with climatic impacts and climate change. Awareness of this diversity is necessary for advancement in the biodeterioration field and necessitates application of new isolation techniques and state-of-art characterization of resulting strains, followed by employing these strains in novel material testing procedures. Isolation, characterization and description of new material-associated strains will support (i) broadening our knowledge of the ecosystem associated with material biodeterioration and (ii) designing new material testing procedures with dominant organisms.

For exposed material surfaces from rock to roof tiles to solar panels, melanized fungal ascomycetes arise as a group of special interest. To understand the diversity of microbial settlers on bare material surfaces, coordinated efforts of biodeterioration scientists, culture collections and “black-fungi-experienced” taxonomy and phylogeny specialists are indispensable.

## Figures and Tables

**Figure 1 life-08-00030-f001:**
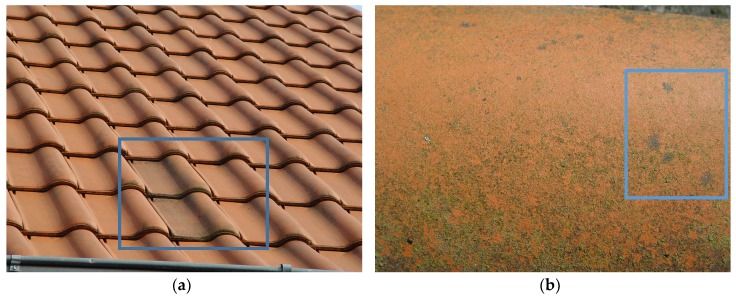
The roof of a residential house with indicated sampled tiles at the moment of sampling in May 2014—after 7.5 years of total exposure and 59 months of exposure beside each other. This area is free from shading by architectural elements and thus photocatalytic activity might occur at least during sun irradiation periods. (**a**) The position of the two sampled photocatalytically active tiles on the control part of the roof was “home” to those tiles for almost five years prior to the sampling. The photocatalytically active tiles have the visibly different subaerial biofilm that is more discoloring. (**b**) Sampled tile surface in a close-up showing dark-pigmented fungal colonies on the elevated surface area—the area where time of wetness is additionally reduced, leading to less pronounced algal growth. Fungal colonies that were sampled with sterile implements to isolate dominant subaerial settlers are indicated by the blue frame.

**Figure 2 life-08-00030-f002:**
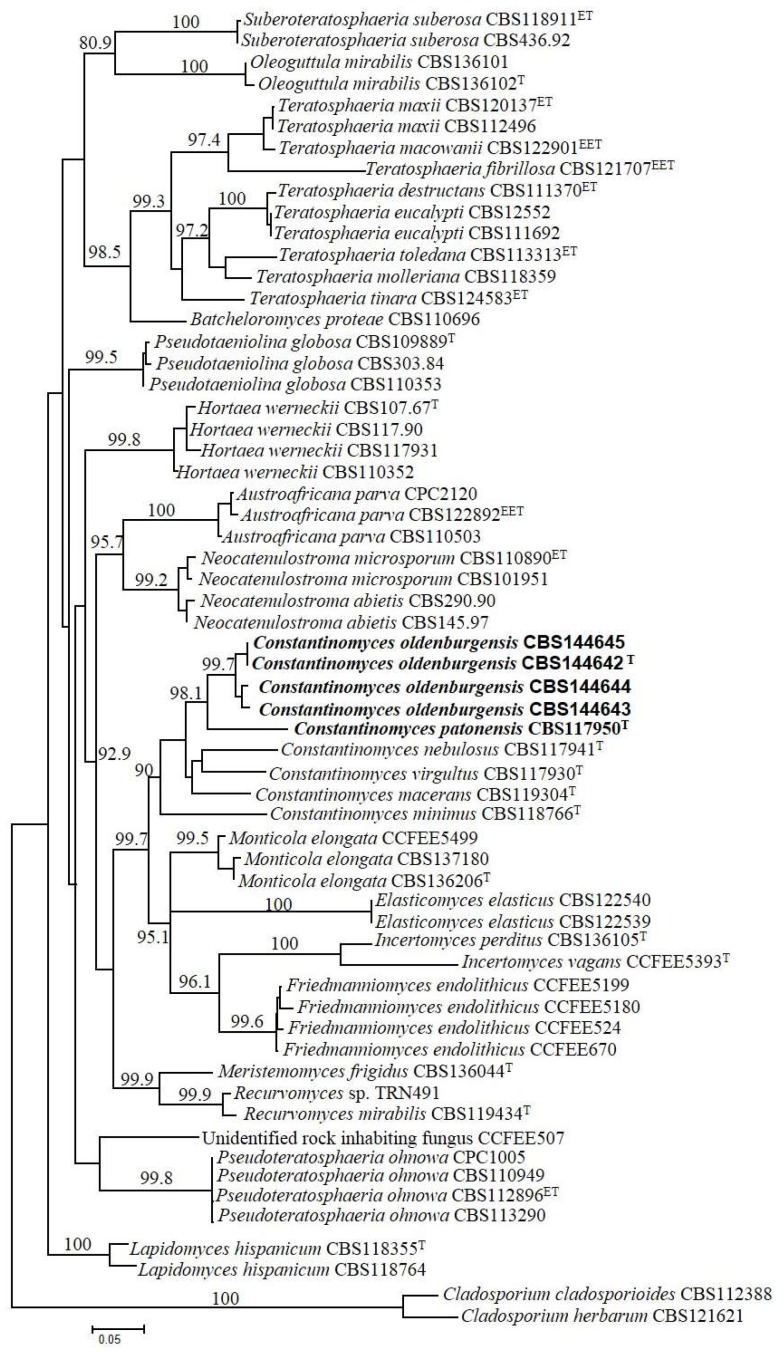
ML 3-genes multilocus phylogeny based on a selection of rock-inhabiting and plant pathogenic fungi in the order Capnodiales showing the phylogenetic placement of the new species described. The tree, based on 59 sequences and 1681 nucleotide positions, has been generated using a GTR + G(4) model calculated using ML in the software MrAIC. Bootstrap values above 80%, calculated from 1000 resampled data sets, are shown.

**Figure 3 life-08-00030-f003:**
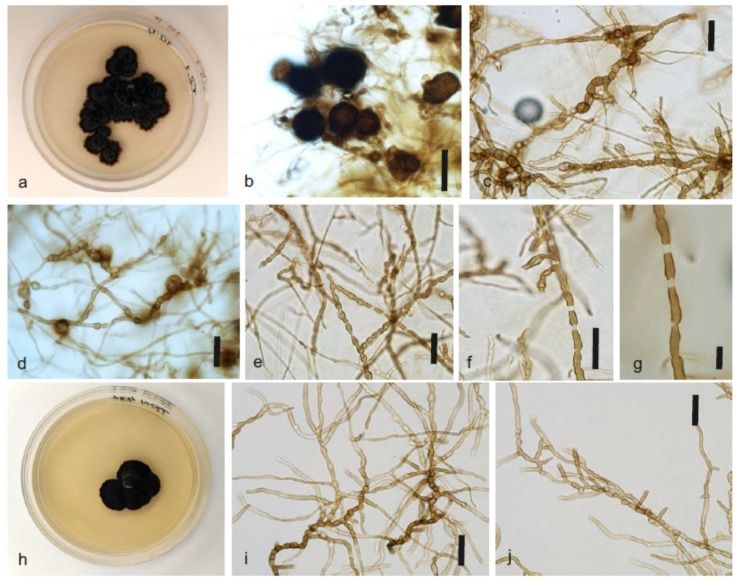
Appearance of the *Constantinomyces oldenburgensis* and *Constantinomyces patonensis*. (**a**–**g**) *Constantinomyces oldenburgensis*, strain T2.1. (**a**) Colony appearance. (**b**) Dark, spherical multicellular bodies. Scale bar 50 µm. (**c**) Irregular toruloid brown hyphae. Scale bar 25 µm. (**d**) Irregular toruloid brown hyphae with development of chlamydospore-like cells. Scale bar 25 µm. (**e**) Micronematous conidiophores and arthroconidia. Scale bar 25 µm. (**f**–**g**) Micronematous conidiophores and arthroconidia, details. Scale bars 10 µm. (**h**–**j**) *Constantinomyces patonensis,* strain TRN431. (**h**) Colony appearance. (**i**–**j**) Irregular toruloid brown hyphae. Scale bars 25 µm.

**Table 1 life-08-00030-t001:** Strains used in this study.

Species	Strain	Origin	Location	GenBank Accession Number		
				ITS	LSU	RPB2
*Austroafricana parva*	CBS122892 = CPC 12421 ^EET^	*Eucalyptus globulus*	Australia	KF901514	KF901832	KF902193
*Austroafricana parva*	CBS110503 = CMW 14459	*Eucalyptus globulus*	Australia	KF901513	KF901831	KF902189
*Austroafricana parva*	CPC2120	*Protea repens*	South Africa	AY260091	-	-
*Batcheloromyces proteae*	CBS110696 = CPC1518 = CPC18701	*Protea cynaroides*	South Africa	-	KF901833	KF902195
*Constantinomyces macerans*	CBS119304 = TRN440 ^T^	Granite, Patones, Central Mountain System	Patones, Spain	AY843139	KF310005	KF310081
*Constantinomyces nebulosus*	CBS117941 = TRN262 ^T^	Granite, Atazar, Central Mountain System	Atazar, Spain	AY843109	KF310014	KF310068
***Constantinomyces oldenburgensis*^T^**	CBS144642 = CCFEE6311 = T2.1	Photocatalytically active roof tiles	Edewecht, Germany	LT976552 *	LT976552 *	LT976526
***Constantinomyces oldenburgensis***	CBS144643 = CCFEE6310 = T2.3	Photocatalytically active roof tiles	Edewecht, Germany	LT976553 *	LT976553 *	LT976527
***Constantinomyces oldenburgensis***	CBS144644 = CCFEE6309 = T2.4	Photocatalytically active roof tiles	Edewecht, Germany	LT976554 *	LT976554 *	LT976528
***Constantinomyces oldenburgensis***	CBS144645 = CCFEE6305 = T2.5	Photocatalytically active roof tiles	Edewecht, Germany	LT976555 *	LT976555 *	LT976529
*Constantinomyces minimus*	CBS118766 = TRN159	Granite, La Cabrera, Central Mountain System	La Cabrera, Spain	AY843066	KF310003	KF310077
***Constantinomyces patonensis*^T^**	CBS117950 = TRN431	Granite, Patones, Central Mountain System	Patones, Spain	AY843129	KF310004	KF310080
*Constantinomyces virgultus*	CBS117930 = TRN79 ^T^	Limestone, Cala Sant Vicenç	Mallorca, Spain	AY559339	GU323964	KF310082
*Elasticomyces elasticus*	CBS122539 = CCFEE5319	*Lecanora* sp.	Antarctica	FJ415475	GU250375	Unpublished
*Elasticomyces elasticus*	CBS122540 = CCFEE5320	*Usnea antarctica*	Antarctica	FJ415476	GU250376	-
*Friedmanniomyces endolithicus*	CCFEE5199	Rock	Antarctica	JN885547	KF310007	KF310093
*Friedmanniomyces endolithicus*	CCFEE5180	Rock	Antarctica	JN885544	GU250367	-
*Friedmanniomyces endolithicus*	CBS119426 = CCFEE670	Rock	Antarctica	JN885542	GU250366	KF310056
*Friedmanniomyces endolithicus*	CBS119427 = CCFEE 524	Rock	Antarctica	JN885541	GU250364	KF310054
*Hortaea werneckii*	CBS107.67 ^T^	Tinea nigra: man	Portugal	AJ238468	EU019270	-
*Hortaea werneckii*	CBS117.90	Salted fish, *Osteoglossum bicirrhosum*	Brazil	AJ238472	-	-
*Hortaea werneckii*	CBS117931 = TRN122	Rock	Cala Sant Vicenç, Mallorca, Spain	AY559357	GU323969	KF310058
*Hortaea werneckii*	CBS110352	Hollow tree	Karthoum	Unpublished	-	-
*Incertomyces perditus*	CBS136105 = CCFEE 5385 ^T^	Rock	Alps, Italy	KF309977	KF310008	KF309977
*Incertomyces vagans*	CCFEE5393 ^T^	Rock	Alps, Italy	KF309964	KF310009	KF309964
*Lapidomyces hispanicus*	CBS118355 = TRN500	Rock	Puebla la Sierra, Spain	AY843182	KF310017	-
*Lapidomyces hispanicus*	CBS118764 = TRN126	Cala Sant Vicenç	Mallorca, Spain	AY559361	KF310016	KF310076
*Meristemomyces frigidus*	CBS136044 = CCFEE5401 ^T^	Rock	Alps, Italy	KF309966	GU250383	KF310105
*Monticola elongata*	CCFEE5499	Rock	Alps, Italy	KF309969	GU250398	KF310065
*Monticola elongata*	CBS137180 = CCFEE 5492	Rock	Alps, Italy	KF309968	KF309994	KF310064
*Monticola elongata*	CBS136206 = CCFEE5394 ^T^	Rock	Alps, Italy	KF309965	KF309995	KF309965
*Neocatenulostroma abietis*	CBS145.97	Sandstone of cathedral	Germany	AJ244265	-	-
*Neocatenulostroma abietis*	CBS290.90	Skin lesion	Netherlands	AJ244267	-	-
*Neocatenulostroma microsporum*	CBS110890 = CPC1832 ^ET^	*Protea cynaroides*	South Africa	AY260097	EU019255	-
*Neocatenulostroma microsporum*	CBS101951 = CPC1960 ^ET^	*Protea cynaroides*	South Africa	AY260097	EU019255	-
*Oleoguttula mirabilis*	CBS136101 = CCFEE5522	Rock	Antarctic Peninsula	KF309972	KF310019	KF310070
*Oleoguttula mirabilis*	CBS136102 = CCFEE5523 ^T^	Rock	Antarctic Peninsula	KF309973	KF310031	Unpublished
*Pseudoteratosphaeria ohnowa*	CBS112896 = CPC1004 ^ET^	*Eucalyptus grandis*	South Africa	KF901620	KF901946	KF902348
*Pseudoteratosphaeria ohnowa*	CPC1005	*Eucalyptus grandis*	South Africa	AF173299	GU214511	-
*Pseudoteratosphaeria ohnowa*	CBS113290 = CMW9102	*Eucalyptus smithii*	South Africa	KF937236	-	KF937270
*Pseudoteratosphaeria ohnowa*	CBS110949 = CPC1006	*Eucalyptus grandis*	South Africa	AY725575	-	-
*Pseudotaeniolina globosa*	CBS109889 ^T^	Rock	Italy	NR136960	EU019283	-
*Pseudotaeniolina globosa*	CBS303.84	Rock	Italy	AJ244268	-	-
*Pseudotaeniolina globosa*	CBS110353	Human aorta at autopsy	Germany	Unpublished	-	-
*Recurvomyces mirabilis*	CBS119434 = CCFEE5264 ^T^	Sandstone	Antarctica	FJ415477	GU250372	KF310059
*Recurvomyces* sp.	CBS117957 = TRN491	Quarzite	Puebla de la Sierra, Spain	AY1843175	-	-
*Suberoteratosphaeria suberosa*	CBS11891 = CPC12085 ^ET^	*Eucalyptus* sp.	Uruguay	KF901786	KF902144	-
*Suberoteratosphaeria suberosa*	CBS436.92 = CPC515 ^ET^	*Eucalyptus dunnii*	Brazil	KF901623	KF901949	KF902404
*Teratosphaeria destructans*	CBS111370 = CPC1368 ^ET^	*Eucalyptus grandis*	Indonesia	KF901574	KF901898	KF902427
*Teratosphaeria eucalypti*	CBS111692 = CMW14910 = CPC1582	*Eucalyptus* sp.	New Zealand	-	KF902119	-
*Teratosphaeria eucalypti*	CPC 12552	*Eucalyptus nitens*	Tasmania	KF901576	KF901900	KF902429
*Teratosphaeria fibrillosa*	CBS121707 = CPC13960 ^EET^	*Protea* sp.	South Africa	KF901728	KF902075	-
*Teratosphaeria macowanii*	CBS122901 = CPC13899 ^EET^	*Protea nitida*	South Africa	KF937241	-	KF937276
*Teratosphaeria maxii*	CBS120137 = CPC12805 ^ET^	*Protea repens*	South Africa	KF937243	-	KF937278
*Teratosphaeria maxii*	CBS112496 = CPC3322	*Protea* sp.	Australia	-	KF937242	KF937277
*Teratosphaeria molleriana*	CBS118359 = CMW11560	*Eucalyptus globulus*	Tasmania	KF901764	KF902120	KF902451
*Teratosphaeria tinara*	CBS124583 = MUCC666 ^ET^	*Corymbia* sp.	Australia	KF901599	KF901923	KF902491
*Teratosphaeria toledana*	CBS113313 = CMW14457 ^ET^	*Eucalyptus* sp	Spain	KF901734	KF902081	KF902492
Unidentified rock inhabiting fungus	CCFEE507	Powdered rocks	Antarctica	Unpublished	-	-

CBS: Westerdijk Fungal Biodiversity Institute, Utrecht, The Netherlands; CCFEE: Culture Collection of Fungi from Extreme Environments, DEB, University of Tuscia, Viterbo, Italy; CMW: Culture collection of the Forestry and Agricultural Biotechnology Institute (FABI) of the University of Pretoria, Pretoria, South Africa; CPC: Collection Pedro Crous; MUCC: Murdoch University Culture Collection, Murdoch, Australia; NZFS: Forest Research Culture Collection, Private Bag 3020, Rotorua, New Zealand; STE-U: Department of Plant Pathology, University of Stellenbosch, South Africa; TRN: C. Ruibal private collection; EET: ex-epitype; ET: ex-type. ITS: internal transcribed Spacers; LSU: Large SubUnit; RPB2: RNA Polymerase II Large Subunit. * ITS plus LSU submitted as single sequence in the GenBank.

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
