# Peer review of "Roof-Inhabiting Cousins of Rock-Inhabiting Fungi: Novel Melanized Microcolonial Fungal Species from Photocatalytically Reactive Subaerial Surfaces"

_life, 2018, doi:10.3390/life8030030_

Round 1

Reviewer 1 Report

The article focuses on four strains of a new fungal species collected on tiles with photocatalytic surface properties, placed on a roof in Germany. Another new species from the same genera, collected on rock in Spain is also described. These two new species are inserted in a phylogenetic analysis. A comparison between tiles treated and not treated with the photocatalytic process is given, showing the different modifications that occur on the surface during 8 years.
This article gives useful taxonomical and phylogenetic information for mycologists, new ecological data on fungi growing on bare substrates as rock and tiles. It also gives important results for industry, on biodegradation of building materials and solar panels surface.
Tree different points should be improved.
First, the photocatalytic treatment used is not described in the Methods section, neither in the description of the new species.
The second point is about the Phylogram. It is not true that the topology of the tree is in accordance with wider recent phylogenetic studies.
The third point concern the taxonomic descriptions: comparison with other species of the genus are missing.

Author Response

We are very grateful to both reviewers for their comments and have carefully addressed all their suggestions. The resulting revision would have been impossible without these valuable and critical impulses that have helped to significantly improve the structure, the style as well as the presentation of the article. On top of all the specific scientific comments, we have taken the expressed style and language concerns very seriously. After all content-related and structural improvements have been done, a native speaker has edited the manuscript thereby eliminating awkward formulations and sentences. We hope that the resulting product meets your expectations and would be happy to answer any question and comment.

Reviewer #1:

The photocatalytic treatment of the tiles was an integral part of a commercially available product. Therefore, no information was ever available to us on the production mode and/or specific composition of its surface layer – and equipped with this knowledge from the very beginning, we aimed at analysis of the observed colonisation pattern and biodiversity differences between two different types of surfaces, in a natural realistic environmental exposure setup. The design of the study was based on the juxtaposed environmental exposure and the careful in-depth mycological analysis of the biological colonisation that has been able to develop on these material surfaces.

Following this comment, we have significantly modified the presentation of the exposure setup in a way that is now clearly stating the rationale of the study. We observe fungal growth as an exposure phenomenon and present the biological indicators (organisms) that have an obvious capacity to grow on this new type of tiles. The comment has stimulated us to modify our experiment description in the introduction, as well as in sections 2.1 and 3.1 accordingly.

Phylogram as well as taxonomic descriptions have been significantly modified in accordance with the critical comments. Detailed changes were multiple and are listed below:

The nature of rocks on which Constantinomyces patonensis is now systematically presented in the table as well as in the text.

The position of Constantinomyces nebulosus and C. macerans was specifically addressed in the revised version by adding an explanatory sentence “Besides, the species C. nebulosus was placed far from C. macerans and in a misguided position in the reference study, whilst all the species hitherto described in the genus Constantinomyces cluster together in our phylogeny which is based on multigene analysis. Lines 244-246

Taxonomic comparison with other species of the genus Constantinomyces have been amended: comparisons between the new species C. oldenburgensis and C. patonensis have been introduced and the differences between these two species and the other species in the genus are clearly presented (line 281-283).

We completely agree with reviewer’s comments 3,4 and addressed this issue by adding C. minimum, the last species in the genus Constantinomyces, into the tree. Besides, Parapenidiella and Piedraia species are distant from the genus Constantinomyces and are apparently close to C. nebulosus in the phylogeny by  Quaedvlieg et al. 2014 just because C.nebulosus was mis-positioned there, quite far from the other Constantinomyces species. In our phylogeny all Constantinomyces species are pooling together and in a congruent position with what was published by Egidi et al., 2014 (the publication where the genus itself and the 4 species were described). Addition of the suggested cluster would bring widening of the cluster around  Pseudoteratosphaeria ohnowa and relatives without adding any useful information for the specific purposes. Catenulostroma species are already in our tree (Neocatenulosroma microsporum and Austroafricana parva group).

Positions in ITS shared by our 4 samples C. oldenbugensis and different in the other species of the genus have been defined. The sentence “The specimens T2.1, T2.3, T2.4, T2.5 here analysed shared 5 positions in ITS sequences that were different in the other close relatives analysed; the positions and substitutions in the analysed specimens/closest relatives are reported as follows: position 26: T/C; position 30: T/C; position 54: T/A or C; position 171-172: TT/AC or AA or AT or TC.” Has been added line 230-233.

We completely agree that reassessment will be necessary when more sequences will become available. As, however, for the moment there is just 1 representative for most of Constatinomyces species, it is not possible to discern characteristic base in a species. We will have to keep isolating for now and searching for the new representative isolates.

Misspellings in the taxa names in the phylogenetic tree change were corrected.

in Table 1: the ITS sequence for Pseudotaeniolina globosa CBS 110353 is unpublished, this information has been added

A reference for the DNA extraction method is added

In the manuscript  “resistance” is defined as “insensitivity to disturbance” and can be used to different types of surface and material characteristics.

Reviewer 2 Report

The paper describes two new extremophile ascomycete species from roof surfaces. The results are interesting, novel and well presented in most parts of the manuscript. However, there are two main weaknesses in the manuscript that should be addressed: First of all in the Methods section, the rationale for moving the two roof tiles to the other roof section is not well explained. It makes this part rather confusing and difficult to read. There are some explanations in the very extensive legend of Figure 1 which are not presented in the main text. It is recommended to re-write the whole section 2.1 with the view of consolidating the information in the text legend with the main text and maybe shortening the figure legend of Fig.1. Maybe the tiles were moved because it was realized later that there is some shading occurring at the original location? The authors should clarify why this step was done (was it part of the original experimental design -- if so why? -- or as a correction to the original design or merely in order to see a contrast?); after all these were the substrates from which the fungi were isolated. The second weakness is the narrative around the fact that there was no (biofilm and fungal) growth on the control tiles. While it is argued throughout the paper that a protection with biocides or with a photocatalytic surface should prevent the growth of biofilms, the results presented in this manuscript seem to indicate that it is the traditional tiles that have less growth. The authors do mention the fact in the discussion, but it should be more emphasized in the respective parts of the manuscript; the results would actually indicate that photocatalytical roof tiles are indeed inferior to traditional ones. Isn't it remarkable that in order to grow these fungi a photocatalytically active surface is necessary and they don't grow on traditional tiles? The manuscript should be thoroughly re-checked for English grammar and expression. Some errors are corrected below, but not throughout the manuscript. Additional comments: Line 24 ...panel (singular) Line 26: delete "the" Line 30: Two different issues are mixed here: resistance development is obvious as a result of biocides, but would it apply to photocatalytic surfaces? Line 31: ...on a subaerial... Line 32: the location of the study should be mentioned in the Abstract Line 34: ... LSU and RBP2 gene ... Line 37-38: Last sentence is not relevant for the Abstract, should be deleted Line 46: ... on desert ... Line 55: ... modifications such as biocides .... Line 57: resistance?, see previous comment for the Abstract. Resistance is what you would expect in the case of biocides, not photocatalytic surfaces. The term "niche: might be a better fit here Line 60: ... with a ... Line 63: ... three-gene multilocus Line 64: ... plant pathogenic ... (correct in other parts of manuscript) Line 99: ... with sterile implements ... (correct throughout manuscript) Line 100: ... to agar media ... Line 103: .... of green algae ... Line 136: ...through the .... Line 138: ....using the ... Line 143: ... using a bootstrap ... Line 147: .... grown on 4% ... Line 148: what does it mean "12 and 35 weeks"? Line 155: ...from 2006-2014 a whole series of changes occurred Section 3.1, first two paragraphs need substantial corrections to English language and style Line 197: As it is frequently ... Line 198: ... the observed strains ... Line 212: ... obtained for ... Line 221: With regard to BLAST Line 229: ... whose sequences .... Line 246: separate Line 257: The observed strains ... Line 275: Chlamydospore (correct everywhere) Line 277: occasionally with thallic-arthric conidia, liberating sparsely Line 281: A similar Line 316: the occurrence of novel ... The Conclusions should be written using passive style (rather than "we will...)

Author Response

We are very grateful to both reviewers for their comments and have carefully addressed all their suggestions. The resulting revision would have been impossible without these valuable and critical impulses that have helped to significantly improve the structure, the style as well as the presentation of the article. On top of all the specific scientific comments, we have taken the expressed style and language concerns very seriously. After all content-related and structural improvements have been done, a native speaker has edited the manuscript thereby eliminating awkward formulations and sentences. We hope that the resulting product meets your expectations and would be happy to answer any question and comment.

Reviewer #2:

Following the attentive comments of the reviewer, the rationale of the environmental exposure of two different tile types is now presented and described in a more straightforward manner. As we were looking only at the soiling-related consequences of exposure to the environment we concentrated on the descriptions of the available facts and the organisms that showed after several years of observation.

The Figure 1 legend and the text in sections 2.1 and 3.1 have been significantly revised and re-written by consolidating the information in the text legend with the main text.

We observed different colonies on photocatalytically active surfaces and were able to isolate different and novel organisms. This information is presented, however, not in the light of the comparative inferiority or superiority, but from the point of view of material-induced biodiversity.

All English grammar and expression comments of the reviewer 2 were gratefully addressed and the text was significantly improved by these changes. After substantial content-related changes of the presentation were complete, the text was revised by a native speaker, a scientist with experience in environmental-related topics.

Round 2

Reviewer 1 Report

The authors has taken into account all comments made by the reviewers.

I only regret that the authors did not give additional information about the photocatalytic treatment of tiles.

Nevertheless, the authors have improved the manuscript that give valid descriptions of the two new species of fungi.

Author Response

Thank you for your effort to improve our article. It was our pleasure to improve the manuscript presentation by including all reviewer's suggestions.

We also deeply regret that the composition of the photocatalytic surface modifications were impossible to obtain and/or check in laboratory experiments.

Reviewer 2 Report

The authors have greatly improved the manuscript and they have addressed all concerns mentioned earlier.

I have only found some minor errors to fix:

Line 100: might occur [separate words]

Line 273: in the 1990s

I suggest to improve:

Line 301:  None of the structures observed in C. oldenburgensis, such as multicellular bodies or chlamydospore-like cells, have been found in C. patonensis.

Author Response

Our sincere gratitude for your effort to improve the manuscript! All your comments were incorporated into the new version of the article.

As now the CBS strain numbers as well as type strains are clear, some minor changes have been done to the tree and the strains list:

type strains for both new species were specified in the table

in the tree the CBS numbers for Constantinomyces oldenburgensis are incorporated for the sake of clarity.